# ADVERSARIAL MASKING: TOWARDS UNDERSTANDING ROBUSTNESS TRADE-OFF FOR GENERALIZATION

## ABSTRACT

Adversarial training is a commonly used technique to improve model robustness against adversarial examples. Despite its success as a defense mechanism, adversarial training often fails to generalize well to unperturbed test data. While previous work assumes it is caused by the discrepancy between robust and non-robust features, in this paper, we introduce *Adversarial Masking*, a new hypothesis that this trade-off is caused by different feature maskings applied. Specifically, the rescaling operation in the batch normalization layer, when combined together with ReLU activation, serves as a feature masking layer to select different features for model training. By carefully manipulating different maskings, a well-balanced trade-off can be achieved between model performance on unperturbed and perturbed data. Built upon this hypothesis, we further propose *Robust Masking* (Rob-Mask), which constructs a unique masking for every specific attack perturbation by learning a set of primary adversarial feature maskings. By incorporating different feature maps after the masking, we can distill better features to help model generalization. Sufficiently, adversarial training can be treated as an effective regularizer to achieve better generalization. Experiments on multiple benchmarks demonstrate that RobMask achieves significant improvement on clean test accuracy compared to strong state-of-the-art baselines.

## 1 INTRODUCTION

Deep neural networks have achieved unprecedented success over a variety of tasks and across different domains. However, studies have shown that neural networks are inherently vulnerable to adversarial examples (Biggio et al., 2013; Szegedy et al., 2014). To enhance model robustness against adversarial examples, adversarial training (Goodfellow et al., 2015; Madry et al., 2018) has become one of the most effective and widely applied defense methods, which employs specific attacking algorithms to generate adversarial examples during training in order to learn robust models.

Albeit effective in countering adversarial examples, adversarial training often suffers from inferior performance on clean data (Zhang et al., 2019; Balaji et al., 2019). This observation has led prior work to extrapolate that a trade-off between robustness and accuracy may be inevitable, particularly for image classification tasks (Zhang et al., 2019; Tsipras et al., 2019). However, Yang et al. (2020) recently suggests that it is possible to learn classifiers both robust and highly accurate on real image data. The current state of adversarial training methods falls short of this prediction, and the discrepancy remains poorly understood.

In this paper, we conduct an in-depth study on understanding the trade-off between robustness and clean accuracy in adversarial training, and introduce *Adversarial Masking*, a new hypothesis stating that a widely used technique, batch normalization (BN), has a significant impact on the trade-off between robustness and natural accuracy. Specifically, we break down BN into normalization and rescaling operations, and find that the rescaling operation has a significant impact on the robustness trade-off while normalization only has marginal influence. Built upon this observation, we hypothesize that adversarial masking (*i.e.*, the combination of the rescaling operation and the follow-up ReLU activation fucntion) acts as a feature masking layer that can magnify or block feature maps to influence the performance of robust or clean generalization. In this hypothesis, different rescaling parameters in BN contribute to different adversarial maskings learned through training. By using a simple linear combination of two adversarial maskings, rather than using robust features learned

by adversarial training (Madry et al., 2018; Ilyas et al., 2019; Zhang et al., 2019), we show that a well-balanced trade-off can be readily achieved.

Based on the Adversarial Masking hypothesis, we further propose RobMask (**Rob**ust **Mask**ing), a new training scheme that learns an adaptive feature masking for different perturbation strengths. We use the learned adaptive feature masking to incorporate different features so that we could improve model generalization with a better robustness trade-off. Specifically, each perturbation strength is encoded as a low-dimensional vector, and we take this vector as input to a learnable linear projection layer together with ReLU activation, to obtain the adversarial masking for the corresponding perturbation strength. Therefore, for different perturbation strengths, we learn different maskings accordingly. By doing so, rather than hurting the performance on clean test data, we use adversarial examples as powerful regularization to boost model generalization. Experiments on multiple benchmarks demonstrate that RobMask achieves not only significantly better natural accuracy, but also a better trade-off between robustness and generalization.

Our contributions are summarized as follows. ($i$) We conduct a detailed analysis to demonstrate that the rescaling operation in batch normalization has a significant impact on the trade-off between robustness and natural accuracy. ($ii$) We introduce Adversarial Masking, a new hypothesis to explain that this trade-off is caused by different feature maskings applied, and different combinations of maskings can lead to different trade-offs. ($iii$) We propose RobMask, a new training scheme to learn an adaptive masking for different perturbation strengths, in order to utilize adversarial examples to boost generalization on clean data. RobMask also achieves a better trade-off between robust and natural accuracy.

## 2 PRELIMINARY AND RELATED WORK

**Adversarial Training**    Since the discovery of the vulnerability of deep neural networks, diverse approaches have been proposed to enhance model adversarial robustness. A natural idea is to iteratively generate adversarial examples, add them back to the training data, and then retrain the model. For example, Goodfellow et al. (2015) uses adversarial examples generated by FGSM to augment the data, and Kurakin et al. (2017) proposes to use a multi-step FGSM to further improve performance. Madry et al. (2018) shows that adversarial training can be formulated as a min-max optimization problem, and proposes PGD attack (similar to multi-step FGSM) to find adversarial examples for each batch. Specifically, for a $K$-class classification problem, let us denote $\mathcal{D} = \{(\boldsymbol{x}_i, y_i)\}_{i=1}^n$ as the set of training samples with $\boldsymbol{x}_i \in \mathbb{R}^d, y_i \in \{1, \ldots, K\}$, where $K$ is the number of classes. Considering a classification model $f_\theta(\boldsymbol{x}) : \mathbb{R}^d \to \Delta^K$ parameterized by $\theta$, where $\Delta^K$ represents a $K$-dimensional simplex, adversarial training can be formulated as:

$$\min_\theta \frac{1}{n} \sum_{i=1}^n \max_{\boldsymbol{x}_i' \in \mathcal{B}_p(\boldsymbol{x}_i, \epsilon)} \ell(f_\theta(\boldsymbol{x}_i'), y_i), \tag{1}$$

where $\mathcal{B}_p(\boldsymbol{x}_i, \epsilon)$ denotes the $\ell_p$-norm ball centered at $\boldsymbol{x}_i$ with radius $\epsilon$, and $\ell(\cdot, \cdot)$ is the cross-entropy loss. The inner maximization problem aims to find an adversarial version of a given data point $\boldsymbol{x}_i$ that yields the highest loss. In general, $\mathcal{B}_p(\boldsymbol{x}_i, \epsilon)$ can be defined based on the threat model, but the $\ell_\infty$ ball is the most popular choice among recent work (Madry et al., 2018; Zhang et al., 2019), which is also adopted in this paper.

For deep neural networks, the inner maximization does not have a closed-form solution. Thus, adversarial training typically uses a gradient-based iterative solver to approximately solve the inner problem. The most commonly used choice is multi-step PGD (Madry et al., 2018) and C&W attack (Carlini & Wagner, 2017). Since then, most defense algorithms (Zhang et al., 2019; Balaji et al., 2019; Wang et al., 2019; Ding et al., 2018; Cheng et al., 2020) are based on a similar min-max framework.

**Trade-off between Robustness and Accuracy**    While effective in improving model robustness, adversarial training is known to bear a performance drop on clean test data. Tsipras et al. (2019) provides a theoretical example of data distribution where any classifier with high test accuracy must also have low adversarial accuracy under $\ell_\infty$ perturbations. They claim that high performance on both accuracy and robustness may be unattainable due to their inherently opposing goals. Zhang et al. (2019) decomposes the robust error as the sum of natural (classification) error and boundary

error, and provides a differentiable upper-bound using the theory of classification-calibrated loss, based on which they further propose TRADES to achieve different trade-offs by tuning the regularization term.

Most recently, Xie et al. (2020) proposes AdvProp to assign another batch normalization for generating adversarial examples, and shows improved performance on clean data in image classification tasks. There are also parallel studies on applying adversarial training to improve clean data performance in language understanding and vision-and-language tasks (Zhu et al., 2019; Jiang et al., 2019; Liu et al., 2020; Gan et al., 2020).

**Batch Normalization**  Batch Normalization is a widely adopted technique that enables faster and more stable training of deep neural networks, below, we provide a brief overview of batch normalization, which paves the way to introduce our method. Specifically, batch normalization (Ioffe & Szegedy, 2015) is proposed to reduce the internal co-variate shift to ease neural network training. Considering a convolutional neural network, we can define the input and output as $I_{b,c,x,y}$ and $O_{b,c,x,y}$, respectively. The dimensions correspond to examples with a batch $b$, channel $c$, and two spatial dimensions $x, y$. A neural network applies the same normalization for all activations in a given channel:

$$O_{b,c,x,y} \leftarrow \gamma \frac{I_{b,c,x,y} - \mu_c}{\sqrt{\sigma_c^2 + \epsilon}} + \beta \ \ \forall b, c, x, y, \tag{2}$$

where $\mu_c = \frac{1}{|\mathcal{B}|} \sum_{b,x,y} I_{b,c,x,y}$ denotes the mean for channel $c$, and $\sigma_c$ denotes the corresponding standard deviation. $\gamma$ and $\beta$ are two learnable parameters for the channel-wise affine transformation, *i.e.*, rescaling operations. $\epsilon$ is a small number to control numerical stability.

## 3 ADVERSARIAL MASKING

### 3.1 BATCH NORMALIZATION ACTS AS ADVERSARIAL MASKING

Ilyas et al. (2019) disentangles adversarial examples as a natural consequence of non-robust features. Specifically, they construct robust features from an adversarial trained "robust model" directly. Therefore, a common belief is that adversarial robustness comes from feature representations learned through adversarial training (Ilyas et al., 2019; Santurkar et al., 2019). An interesting question we would like to ask is: can we learn robust features from a vanilla standard-trained model, or, can we obtain non-robust features from an adversarial trained "robust model"?

To answer this question, we design the following experiments. We first train a ResNet-18 model with standard and adversarial training, then finetune the networks by allowing only batch normalization (BN) to be changed while freezing other parameters. Specifically, we finetune BN in a standard trained model using adversarial training, and finetune BN in an adversarial trained model with standard training, respectively. Results are summarized in Table 1. Given a standard trained model, by only performing adversarial finetuning of the BN layers, the resulting model can already achieve a reasonably good robust accuracy $26.51\%$ (the 1st block). Similarly, given an adversarial trained model, by only performing standard finetuning of the BN layers, the clean accuracy of the model increases significantly from $78.47\%$ to $86.96\%$ (the 2nd block).

| Method | Clean Acc. | Robust Acc. |
|---|---|---|
| Standard Training | 91.97% | 0.0% |
| + Adv. Finetuning of BN | 53.96% | 26.51% |
| Adv. Training | 78.47% | 48.67% |
| + Standard Finetuning of BN | 86.96% | 5.83% |

Table 1: Clean and robust accuracy of ResNet-18 models trained under different settings on CIFAR-10. All robust accuracy results are obtained using the $\epsilon = 8/255 \ \ell_\infty$ ball. (BN: Batch Normalization)

This experiment demonstrates that we can control the trade-off between clean and robust errors by only tuning the BN layer, so here comes a natural question: which part in BN has contributed to this performance trade-off? To investigate this, we take a step further to check the difference of every parameter used in BN. In particular, we check the first BN layer after the first convolution layer.

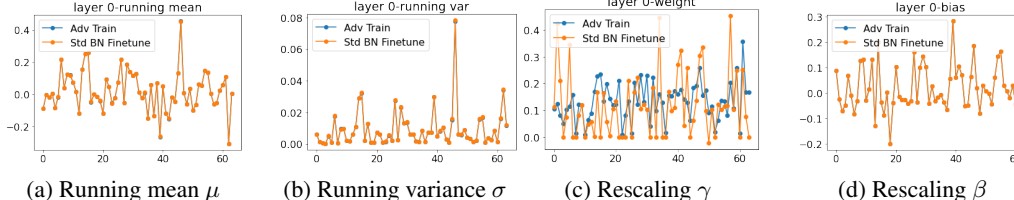

| (a) Running mean $\mu$ | (b) Running variance $\sigma$ | (c) Rescaling $\gamma$ | (d) Rescaling $\beta$ |

Figure 1: Batch statistics in the first batch normalization (BN) layer of an adversarial trained ResNet18 model on CIFAR10, with and without further standard fine-tuning of BN (orange and blue lines, respectively). The running mean $\mu$ and variance $\sigma$, as well as the rescaling shift parameter $\beta$ are almost the same (overlapped in the figure), while the rescaling weight $\gamma$ has a significant difference, which has a notable contribution to the clean and robustness trade-off.

| Model | $p$ | 1.0 | 0.8 | 0.6 | 0.4 | 0.2 | 0.0 |
|---|---|---|---|---|---|---|---|
| Adv. Training with and | Clean Acc. | 78.47% | 81.16% | 82.8% | 84.66% | 86.06% | 86.96% |
| w/o Std. Finetuning of BN | Robust Acc. | 48.67% | 44.88% | 37.2% | 24.79% | 12.85% | 5.83% |
| Std. Training with and | Clean Acc. | 91.97% | 88.66% | 74.87% | 58.76% | 53.89% | 53.96% |
| w/o Adv. Finetuning of BN | Robust Acc. | 0.0% | 0.0% | 0.12% | 5.73% | 19.93% | 26.51% |

Table 2: The clean and robust accuracy using different combination coefficient $p$ on CIFAR-10 with ResNet-18. The 1st block uses adversarial trained models with and without further standard fine-tuning of batch normalization (BN). The 2nd block uses standard trained models with and without further adversarial finetuning of BN. All robust accuracies are obtained using $\epsilon = 8/255 \, \ell_\infty$ ball.

As well known, BN uses a running average of the mean and variance during testing. Figure 1a and 1b illustrate the difference on the running mean $\mu$ and running variance $\sigma$. The batch statistics with and without further standard finetuning of BN (under the setting in the 2nd block of Table 1) are nearly identical across all the dimensions. Figure 1c and 1d plot the learned rescaling parameters $\gamma$ and $\beta$. We can clearly see that the fine-tuned rescaling parameter $\gamma$ is completely different from the original one, with $\beta$ unchanged, indicating that $\gamma$ has a significant impact on the clean and robust trade-off while still performing similar normalization on both sides.

On the other hand, Rectified Linear Unit (ReLU) (Agarap, 2018) is the most commonly used activation function in deep neural networks. The function returns 0 if it receives any negative input, and for any positive value $x$ it returns that value back (*i.e.*, $f(x) = \max(0, x)$). During the rescaling operation, $\gamma$ would magnify or shrink the magnitude of feature maps. Together with $\beta$, after ReLU activation, features that become negative will be blocked as 0. By combining the rescaling operation with the following ReLU activation function, the resulting layer can be viewed as a masking layer to magnify or block the features maps from the convolution layer (see Figure 2(a) for illustration). To further validate this, we plot the feature maps after ReLU activation functions in Figure 4 in the Appendix. Some feature maps are blocked after finetuning BN (completely black when all pixels are set to 0) as well as some are magnified significantly. We term the above observation as *Adversarial Masking*, and hypothesize that this leads to the trade-off between robustness and natural accuracy.

### 3.2 Controlling Robustness Trade-off via Adversarial Masking

The above analysis suggests that, rather than feature representations, rescaling in the BN layer together with ReLU activation function serves as a masking layer for selecting different feature combinations that can achieve different performance trade-offs between clean and perturbed test sets. From this hypothesis, a different combination of BN together with ReLU can be regarded as a different adversarial masking. With such a masking, we can readily achieve a series of trade-offs, without the need of training the model from scratch again, which is the case for conventional adversarial training. In the following experiment, we use a simple linear combination of two learned adversarial maskings to achieve this trade-off, and empirically, we observe that this simple design is sufficient. Specifically, denote $(\gamma, \beta)$ and $(\gamma', \beta')$ as two learned adversarial maskings (*i.e.*, the learned rescal-

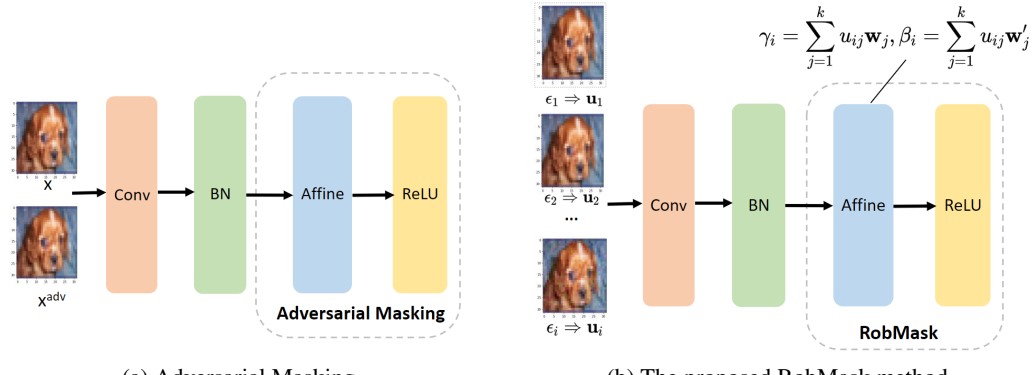

(a) Adversarial Masking.       (b) The proposed RobMask method.

Figure 2: Illustration of (a) Adversarial Masking hypothesis, and (b) RobMask method for improving the generalization performance. Instead of just using a single masking for both clean and adversarial examples, we use the linear combination of $k$ primary rescaling parameters $\{w_j\}_{j=1}^k$ and $\{w'_j\}_{j=1}^k$ to incorporate different perturbation strength $\epsilon_i$.

ing parameters in the BN layer), and we have $\hat{\gamma} = p\gamma + (1-p)\gamma'$ and $\hat{\beta} = p\beta + (1-p)\beta'$. We then use the new adversarial masking $(\hat{\gamma}, \hat{\beta})$ for evaluation. Table 2 shows that different clean and robust accuracies can be readily achieved by selecting different $p$ values.

Previous work (Zhang et al., 2019) uses regularization hyperparameter $\lambda$ to balance between clean error and robust error. By tunning $\lambda$, they could achieve different robustness trade-offs. However, it takes enormous time and effort to retrain the model from the scratch. Instead, this finding inspires us that we can just store one model and employ a series of learned adversarial maskings to control the robustness trade-off at real time.

## 4 IMPROVING MODEL GENERALIZATION VIA ROBMASK

As mentioned in Section 3.2, different trade-offs can be achieved by linearly combining two pre-trained batch normalization layers. However, this may not be ideal due to several deficiencies. First, the "clean" mask learned by fine-tuning is not distilled and may partially override the mask learned from adversarial examples, leading to a sub-optimal solution. Second, since every perturbation strength tends to have a different masking, if we only utilize one perturbation strength, we lose all the other maskings generated by the perturbation strength in-between. Third, it requires a careful selection of what the maximum perturbation strength is. In the extreme case, if a sample is perturbed to the decision boundary, the learned adversarial mask might be completely meaningless. Or, if the chosen perturbation strength is too small, there will not be enough regularization for improving generalization. To address these issues, we propose RobMask (**Rob**ust **Mask**ing), a new framework that aims to actively learn the adversarial masking to boost generalization performance.

Specifically, we propose to incorporate different perturbation strengths for model training, instead of just one. Note that we could treat $\epsilon = 0$ for the unperturbed data. Then, a straightforward way is to just learn a set of $\gamma_i, \beta_i$ independently for every perturbation strength $\epsilon_i$. However, due to the limited number of sampled perturbation strengths, each $\gamma_i$ could have poor generalization due to over-fitting. At the same time, it loses the correlation between all the generated maskings. Instead, we need to jointly learn the corresponding maskings simultaneously.

To this end, we assume that every rescaling parameter $\gamma_i$ can be well-approximated by a linear combination of $k$ basic rescaling parameters $\{w_j\}_{j=1}^k$, where $k$ is a small number. By encoding perturbation strength $\epsilon_i$ into a $k$-dimensional vector $u_i$, we can linearly combine $w_j$ using $u_i$ to obtain a rescaling parameter for strength $\epsilon_i$ as: $\gamma_i = \sum_{j=1}^k u_{ij} w_j$. Take $k = 2$ as an example: we can encode $u_0 = [1.0\ 0.0]^T$ for $\epsilon = 0$ and $u_{\epsilon_{max}} = [0.0\ 1.0]^T$ for $\epsilon_i = \epsilon_{max}$, respectively. Naturally, the intermediate perturbation strength $\epsilon_i = p_i \cdot \epsilon_{max}$ can be encoded as $u_i = (1 -$

---

**Algorithm 1** The proposed RobMask method for improving model generalization.

---

**Input:** Training dataset $\{\boldsymbol{x}_i, y_i\}_{i=1}^n$, perturbation upper bound $\epsilon_{max}$.
**for** epoch$= 1, \ldots, N$ **do**
   **for** $i = 1, \ldots, B$ **do**
      Sample a random number $p$ from 0 to 1
      $\epsilon_i \leftarrow p \cdot \epsilon_{max}$   # obtain the current perturbation strength in this mini-batch
      $\boldsymbol{u}_i \leftarrow \text{Encode}(\epsilon_i)$   # encoder the perturbation strength as a vector
      $\boldsymbol{\delta}_i \leftarrow 0$
      **for** $j = 1, \ldots, m$ **do**
         $\boldsymbol{\delta}_i \leftarrow \boldsymbol{\delta}_i + \alpha \cdot \text{sign}(\nabla_{\boldsymbol{\delta}}\ell(f_\theta(\boldsymbol{x}_i + \boldsymbol{\delta}_i), y_i)$   # PGD adversarial attack
         $\boldsymbol{\delta}_i \leftarrow \max(\min(\boldsymbol{\delta}_i, \epsilon_i), -\epsilon_i)$
      **end for**
      $\gamma_i \leftarrow \sum_{j=1}^k u_{ij}\boldsymbol{w}_j, \beta_i \leftarrow \sum_{j=1}^k u_{ij}\boldsymbol{w}'_j$   # the rescaling parameters in BN
      $\mathbf{W} \leftarrow \mathbf{W} - \eta_1 \cdot \nabla_{\mathbf{W}}\ell(f_\theta(\boldsymbol{x}_i + \boldsymbol{\delta}_i, \gamma_i, \beta_i), y_i)$   # update $\mathbf{W}$ and $\mathbf{W}'$
      $\mathbf{W}' \leftarrow \mathbf{W}' - \eta_2 \cdot \nabla_{\mathbf{W}'}\ell(f_\theta(\boldsymbol{x}_i + \boldsymbol{\delta}_i, \gamma_i, \beta_i), y_i)$
      $\theta \leftarrow \theta - \eta_3 \cdot \nabla_\theta\ell(f_\theta(\boldsymbol{x}_i + \boldsymbol{\delta}_i, \gamma_i, \beta_i), y_i)$   # update neural network parameters $\theta$
   **end for**
**end for**
**return** $\theta, \mathbf{W}, \mathbf{W}'$

---

$p_i)\boldsymbol{u}_0 + p_i\boldsymbol{u}_{\epsilon_{max}}$. Therefore, instead of learning $\gamma_i$ separately for every $\epsilon_i$, we learn a low-rank matrix $\mathbf{W} = [\boldsymbol{w}_1, \boldsymbol{w}_2, \ldots, \boldsymbol{w}_k]$ to incorporate different perturbation strengths and learn a series of maskings. Similarly, we learn another matrix $\mathbf{W}'$ for $\beta_i$.

During training, in every iteration, with a randomly selected perturbation strength $\epsilon_i = p_i \cdot \epsilon_{max}$, we first generate a mini-batch of adversarial examples by conducting PGD attacks. Then, we learn the rescaling parameter of BN by using a low-rank linear layer ($\mathbf{W}$ and $\mathbf{W}'$) and encoded attack strength $\boldsymbol{u}_i$. Finally, we minimize the total loss using stochastic gradient descent (SGD) to update model parameters. Detailed algorithm is summarized in Algorithm 1.

**Connection with AdvProp** Xie et al. (2020) hypothesizes that the performance degradation on unperturbed test dataset is mainly caused by the distribution mismatch between adversarial examples and clean images. They propose AdvProp to assign an auxiliary batch normalization for adversarial examples, and show that adversarial examples can be useful to achieve better performance on clean test data. However, as shown in Figure 1, the running mean and variance are kept the same after fine-tuning. We argue that the improved model generalization is realized by a different adversarial mask learned by auxiliary batch normalization in the AdvProp procedure.

Although we utilize adversarial training to boost generalization as well, our approach has clear differences. First, in AdvProp, there is no connection between the traditional and auxiliary batch normalization (BN). The traditional BN only obtains inputs from clean data, and the auxiliary BN only obtains inputs from adversarial examples. This type of disentanglement would completely separate different masks, which violates the reality that some masks can be useful for both clean and robust performance. Second, AdvProp has to designate a perturbation strength that should not be too large or too small, which is difficult to tune in practice. Also, the proposed RobMask method is more general than Advprop, and AdvProp can be considered as one special case of RobMask when we set the linear layer rank $k$ to 2 and freeze $p = 1$ in the whole training process.

## 5 EXPERIMENTS

In this section, we conduct experiments to show that RobMask can successfully improve generalization performance. We also provide additional robustness evaluation for completeness.

### 5.1 EXPERIMENTAL SETUP

**Datasets and Model Architectures** We use two popular datasets CIFAR-10 and CIFAR-100 (Krizhevsky et al., 2009) for experiments. For model architectures, we use the popular

ResNet (He et al., 2016a) family including Preact ResNet (He et al., 2016b), ResNeXt (Xie et al., 2016) and the recent well-performed DenseNet (Huang et al., 2016).

**Baselines** We compare RobMask with two baselines: ($i$) AdvProp: Dual batch normalization (Xie et al., 2020), where different batch normalizations are used for clean and adversarial examples during training; and ($ii$) BN: Standard training with normal batch normalization enabled. Note that as our main goal is to improve the generalization performance instead of robust test accuracy, we do not compare against standard adversarial training methods, as they are reported to largely decrease generalization performance (Madry et al., 2018; Zhang et al., 2019; Balaji et al., 2019).

**Implementation Details** For CIFAR-10 and CIFAR-100, we set the number of iterations in adversarial attack to 7 for all the methods during training. All PGD attacks are non-targeted attacks with random initialization. We set the PGD attack strength $\epsilon = 8/255$ with cross-entropy (CE) loss and the step-size to $\epsilon/5$. All models are trained using SGD with momentum 0.9, weight decay $5 \times 10^{-4}$. We use cosine learning rate scheduling with initial learning rate $\gamma_1 = \gamma_2 = \gamma_3 = 0.1$. To have a fair comparison, for RobMask, we set $k = 2$ and $\epsilon_{max} = 8/255$ in all our experiments, which has the same number of model parameters and same regularization strength as AdvProp. All our experiments are implemented in Pytorch. Code will be released upon acceptance.

## 5.2 EXPERIMENTAL RESULTS

**Generalization** Table 3 summarize the results of all the evaluated methods on CIFAR-10 and CIFAR-100. Across all the tested model architectures, RobMask shows a significant improvement over both normal batch normalization (BN) and AdvProp. Specifically, as shown in Table 3, for 100-epochs training on CIFAR-10, RobMask achieves around $1.5\%$ test accuracy improvement over BN and $0.8\%$ over AdvProp, respectively. Similar improvements can also be observed on the CIFAR-100 dataset. Further, when comparing results between Table 3, we observe that RobMask also leads to faster convergence: 20-epochs training using RobMask leads to results that are comparable to 100-epochs training using BN.

Additionally, we add the large-scale dataset ImageNet into the comparison. Table 4 summarize the results with ResNet-18 on ImageNet datasets. It could be clearly seen that while Advprop has a very limited improvement on ResNet-18, RobMask has around 0.4 percent improvement, which further shows RobMask's effectiveness on the generalization.

**Robustness Evaluation** In addition to improved generalization performance, our method can also achieve a better robust and natural accuracy trade-off over adversarial training. For CIFAR-10 and CIFAR-100, we evaluate all the methods under the white-box $\epsilon = 8/255$ $\ell_\infty$-norm bounded non-targeted PGD attack. Specifically, we use 100-step PGD with step size $\epsilon/5$) that is equipped with random start. Moreover, to further verify the robust accuracy achieved, we use the Autoattack(Croce & Hein, 2020) to evaluate the performance. Note that, when $\epsilon = 0$, robust accuracy is reduced to the test accuracy of unperturbed (natural) test samples, *i.e,* clean accuracy. Results are summarized in Table 5 and 6. RobMask clearly outperforms other methods among $\epsilon$ from 0 to 6/255. However, RobMask performs slightly worse on $\epsilon = 8/255$. It is because both AdvProp and Adversarial training models are trained with adversarial examples generated with $\epsilon = 8/255$, while our methods use a random perturbation where $\epsilon_{max} = 8/255$. That is, we use a weaker perturbation strength compared to both AdvProp and Adv train.

To achieve a better result on $\epsilon = 8/255$, we relax the max epsilon constraint from $8/255$ to $10/255$. From Table 6, we could see that the clean performance drop slightly with a increasing robust accuracy on $\epsilon >= 6/255$ so that we now could achieve a better robust accuracy on $\epsilon = 8/255$. Even with the slight degrading performance on clean accuracy, RobMask achieves a better adversarial robustness trade-off over other methods.

**Importance of low-rank matrix:** We conduct an ablation study on DenseNet-121 over CIFAR-10 to investigate the importance of using a low-rank matrix to incorporate multiple perturbation strengths. Here, we extend AdvProp to use a randomly selected strength $\epsilon_i = p_i \cdot \epsilon_{max}$ to generate adversarial examples, and then feed into auxiliary batch normalization. Advprop can also be generalized using multiple auxiliary BNs when given multiple perturbation strengths. In the experiments, instead of an auxiliary batch normalization for adversarial examples generated by $\epsilon = 8/255$, we also give another batch normalization for $\epsilon = 4/255$ adversarial examples. Table 7 shows Advprop

| Model | #Epochs | CIFAR-10 | | | CIFAR-100 | | |
|---|---|---|---|---|---|---|---|
| | | BN | AdvProp | RobMask | BN | AdvProp | RobMask |
| ResNet-18 | 20 | 92.59% | 93.45% | **94.64%** | 75.88% | 76.15% | **77.61%** |
| | 100 | 94.87% | 95.3% | **96.10%** | 77.7% | 76.82% | **78.77%** |
| DenseNet-121 | 20 | 93.26% | 94.61% | **95.30%** | 74.79% | 73.61% | **75.11%** |
| | 100 | 94.71% | 94.61% | **96.47%** | 77.63% | 76.88% | **80.18%** |
| Preact-18 | 20 | 91.93% | 92.55% | **94.04%** | 70.79% | 72.71% | **73.59%** |
| | 100 | 94.37% | 95.33% | **95.97%** | 76.14% | 76.87% | **78.19%** |
| ResNeXt-29 | 20 | 93.12% | 93.37% | **95.03%** | 74.26% | 72.18% | **74.65%** |
| | 100 | 95.15% | 95.29% | **96.06%** | 78.60% | 76.35% | **79.54%** |

Table 3: Comparison on CIFAR-10/100 over ResNet-18, DenseNet-121, Preact-18, and ResNeXt-29. Models are trained for 20 and 100 epochs using normal Batch Normalization (BN), AdvProp and our RobMask. RobMask shows a significant performance improvement on all model architectures. Also, RobMask trained with 20 epochs achieves a comparable performance to 100-epoch training using BN and AdvProp.

| Model | ImageNet | | |
|---|---|---|---|
| | BN | AdvProp | RobMask |
| ResNet-18 | 69.76% | 69.79% | **70.14%** |

Table 4: Comparison on ImageNet over ResNet-18. Models are trained for 105 epochs using normal Batch Normalization (BN), AdvProp and our RobMask. RobMask shows a significant performance improvement on all model architectures.

| $\epsilon$ | 0 | 2/255 | 4/255 | 6/255 | 8/255 |
|---|---|---|---|---|---|
| Adv. Training | 78.86% | 72.61% | 65.27% | 57.15% | **47.97%** |
| AdvProp | 85.98% | 78.34% | 69.24% | 57.61% | 46.04% |
| RobMask | **89.99%** | **82.91%** | **72.23%** | **58.63%** | 44.50% |

Table 5: Robust accuracy under different levels of PGD $\ell_\infty$ attacks on CIFAR-10 with ResNet-18 architecture. RobMask clearly outperforms AdvProp and standard adversarial training in all the test perturbation strengths except $\epsilon = 8/255$ on which AdvProp and standard adversarial training are trained.

| $\epsilon$ | 0 | 2/255 | 4/255 | 6/255 | 8/255 |
|---|---|---|---|---|---|
| Adv. Training | 78.86% | 70.99% | 62.94% | 53.83% | 44.66% |
| AdvProp | 85.98% | 77.73% | 67.57% | 55.36% | 43.13% |
| RobMask | **89.99%** | **81.87%** | **70.99%** | 55.9% | 41.7% |
| RobMask $\epsilon = 10/255$ | 86.5% | 78.47% | 68.08% | **56.89%** | **44.69%** |

Table 6: Robust accuracy under different levels of $\ell_\infty$ Autoattacks on CIFAR-10 with ResNet-18 architecture. RobMask clearly outperforms AdvProp and standard adversarial training in all the test perturbation strengths except $\epsilon = 8/255$ on which AdvProp and standard adversarial training are trained.

has degenerated performance when using random perturbation strength. When adding more auxiliary batch normalization, the performance improves slightly, also observed in Xie et al. (2020). However, RobMask still significantly outperforms AdvProp variations.

**Training curve**: In Figure 3, we plot the training curves for both RobMask and Advprop for CIFAR-10 datasets on DenseNet-121. RobMask clearly outperforms Advprop during the whole training process. Also, RobMask will not introduce additional training overhead cost than AdvProp. Since both RobMask and AdvProp use two forward and one backward pass, they have almost identical training time per epoch. For example, in DenseNet-121, it takes around 870 seconds for Advprop and around 890 seconds for RobMask. i.e., our model did not take much longer to train.

| Method | Clean Acc. |
|---|---|
| AdvProp | 94.61% |
| with random perturbation strength | 94.50% |
| with 2 auxiliary BN | 95.10 % |
| RobMask | **96.47%** |

Table 7: Comparison on CIFAR-10 over DenseNet-121 on AdvProp and its extensions. Models are trained for 100 epochs.

## 6 CONCLUSIONS

In this paper, we analyze the impact of batch normalization (BN) on adversarial robustness, and show that the rescaling operations in BN has a strong impact on the clean and robustness trade-off. We then formalize the rescaling operations together with ReLU activations as an adversarial mask, and show that a simple linear combination of two adversarial maskings can be utilized directly to achieve different performance trade-offs. Inspired by these findings, we propose RobMask, an active adversarial mask learning method that is designed to achieve better generalization performance. The success of RobMask indicates that adversarial training can serve as a strong regularizer, instead of a performance killer, for training better models.

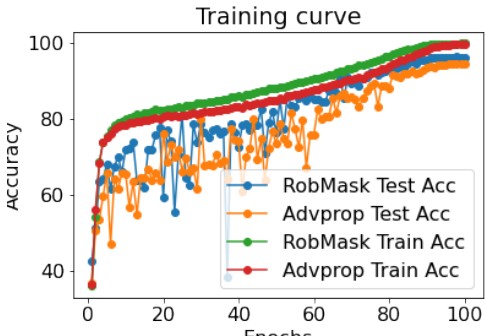

Figure 3: Training curve on DenseNet-121

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

# A  APPENDIX

## A.1  MULTIPLE RUNS

We run RobMask for three times with different random seed and present the mean in Table 8. We report the more detailed result in Table 8:

| Model | #Epochs | CIFAR-10 | CIFAR-100 |
|---|---|---|---|
| ResNet-18 | 20 | 94.54±0.22 | 77.41± 0.20% |
|  | 100 | 95.90±0.24 | 8.59±0.18% |
| DenseNet-121 | 20 | 95.10± 0.20% | 75.94± 0.17% |
|  | 100 | 96.29± 0.18% | 79.99± 0.18% |
| Preact-18 | 20 | 94.92± 0.12% | 73.43± 0.16% |
|  | 100 | 95.83± 0.14% | 78.01±0.18% |
| ResNeXt-29 | 20 | 94.83± 0.21% | 74.43± 0.22% |
|  | 100 | 96.84± 0.22% | 79.31± 0.23% |

Table 8: RobMask results on CIFAR-10/100 over ResNet-18, DenseNet-121, Preact-18, and ResNeXt-29. Models are trained for 20 and 100 epochs.

## A.2  COMPARISON ON BATCH NORMALIZATIONS

In this section, we extend our experiment in Figure 1 to show the batch statistics in the deeper layers across the neural networks. It clearly shows that the rescaling weight has more effect than other parameters in the batch normalization. To be noted, since the deep layer's mean and variance would be affected by the shallow layers' rescaling weight parameter, the result on the deeper layer couldn't disentangle the effect between normalization and rescaling because it is mixed.

|  | Mean | Variance | Weight | Bias |
|---|---|---|---|---|
| Layer 0 | 1.0 | 1.0 | 0.7620 | 1.0 |
| Layer 1 | 0.9842 | 0.9718 | 0.7883 | 1.0 |
| Layer 2 | 0.9530 | 0.9199 | 0.7544 | 1.0 |
| Layer 3 | 0.9743 | 0.9691 | 0.8594 | 1.0 |
| Layer 4 | 0.8894 | 0.9340 | 0.8126 | 1.0 |
| Layer 5 | 0.9555 | 0.9516 | 0.8813 | 1.0 |
| Layer 6 | 0.9853 | 0.9452 | 0.7141 | 1.0 |
| Layer 7 | 0.9554 | 0.9169 | 0.8609 | 1.0 |
| Layer 8 | 0.9903 | 0.9646 | 0.8961 | 1.0 |
| Layer 9 | 0.9635 | 0.9755 | 0.8046 | 1.0 |
| Layer 10 | 0.9823 | 0.9522 | 0.9396 | 1.0 |
| Layer 11 | 0.9823 | 0.9769 | 0.7906 | 1.0 |
| Layer 12 | 0.9753 | 0.9593 | 0.7839 | 1.0 |
| Layer 13 | 0.9914 | 0.9874 | 0.8891 | 1.0 |
| Layer 14 | 0.9699 | 0.9898 | 0.6593 | 1.0 |
| Layer 15 | 0.9902 | 0.9870 | 0.8605 | 1.0 |
| Layer 16 | 0.9603 | 0.9889 | 0.7423 | 1.0 |
| Layer 17 | 0.9736 | 0.9742 | 0.6809 | 1.0 |
| Layer 18 | 0.9772 | 0.9838 | 0.9573 | 1.0 |

Table 9: Cosine similarity on under every batch normalization layer under standard fine-tuned training on adversarial trained model

## A.3  ADVERSARIAL MASKING VISULIAZATION

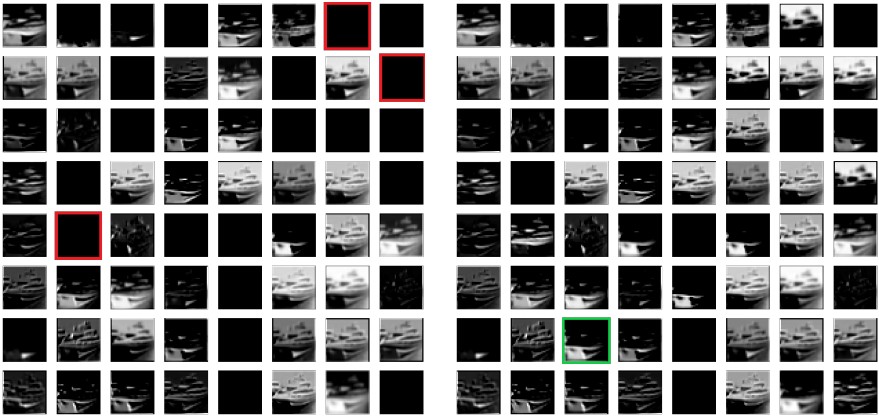

(a) Adversarial training with further standard finetuning of BN

(b) Adversarial training

Figure 4: Illustration of the Adversarial Masking effect. We mark several feature maps (red and green boxes) are blocked out or magnified when comparing (a) and (b), which can be viewed as a selection mask on "non-robust" and "robust" features.

