# OpenReview forum: "Adversarial Masking: Towards Understanding Robustness Trade-off for Generalization"
_ICLR.cc/2021/Conference — Reject_

### Official Review · AnonReviewer2 · 2020-10-28
**Interesting approach, but needs further analysis**

**Rating:** 5
**Confidence:** 5

**Review:**

Summary:
The paper investigates the role of Batch Normalization (BN) in the generalization of deep networks, and its impact in the trade-off between clean and robust accuracy in adversarially trained networks. The authors demonstrate that the rescaling operations in BN when considered in conjunction with the ReLU activation, serve as a feature masking operation. Based on these observations, the authors propose RobMask, which uses a linear combination of such rescaling operations to achieve improved generalization in deep networks.


Pros:
1) Interesting observation, and highlights the needs to re-examine techniques from the standard training paradigm that are directly used in adversarial training of deep networks
2) Easy to integrate with any network architecture that already uses Batch Normalization
3) Demonstrates enhanced standard performance over different network architectures and datasets
4) Achieves improved trade-off between clean and robust accuracy for adversarially trained networks on smaller constraint sets (L-infinity eps = 2/255 to 6/255)

Cons:
1) The paper lacks novelty from the standpoint that very similar observations were made by Xie et al. [1].
2) Further, the primary results are demonstrated for the case of k=2, using a linear combination of Batch Normalization parameters obtained for normal and adversarial training, which represents only a minor change from the algorithm proposed in [1].
3) While the method itself is not difficult to understand, it is still unclear why it helps strike a better balance for the accuracy-robustness trade-off in adversarially trained networks. Could the authors provide an intuitive or theoretical explanation for the same?
4) The adversarial training of larger networks such as DenseNet-121 can be highly computationally intensive, requiring almost an additional order of magnitude in training time. Thus the performance comparison made in Table-3 at the 20th epoch for models trained using normal training and RobMask is unfair, since RobMask uses 7-step adversarial training. Perhaps a better metric to consider would be the standard performance obtained after a fixed training time.
5) The clean accuracy shown in Table-1 and Table-3 for the adversarially trained ResNet-18 model on CIFAR-10 is quite low (78%), compared to standard values reported in [2,3], which is often around 82%.
6) To quote from the last para of Section4: “Also, the proposed RobMask method is more general than Advprop, and AdvProp can be considered as one special case of RobMask when we set the linear layer rank k to 2 and freeze p = 1 in the whole training process.” Thus, could the authors provide additional results for k=3 or k=4? This is quite important to set apart the proposed method from AdvProp, which is highly similar, as final results are only reported with k=2 in Table-3.
7) In test time, it is not immediately clear what choice of $u_i$ should be used, since it determines the effective batch normalisation parameters that are used. Could the authors clarify the exact BN parameters used in final evaluation on clean and adversarial samples?
8) When k is set to a value larger than 2, it is not immediately clear what the different $w_k$’s would represent, since the 2-dimensional $u_i$’s already encode information about different $\ell_\infty$ constraints. Could the authors clarify this? Further, it would be beneficial to the reader to include an example with k=3, along similar lines to what is presented in Section 4.
9) Evaluation on PGD-100 step attack alone is not sufficient, as with stronger attacks such as MultiTargeted attack [4] and AutoAttack [5], the difference in adversarial accuracies might be much lower for eps=2/255 to 6/255, as reported in Table-3.
10) Since the authors say that the proposed methods does not achieve better adversarial accuracy for eps=8/255 in Table-4 due to the sampling of p, could the authors show results where the maximum constraint is set to 10/255 to show an improvement for the standard evaluation setting of 8/255? This would greatly help in the comparison of different methods, particularly the clean accuracy of different models.

[1] Xie, C., Tan, M., Gong, B., Wang, J., Yuille, A. L., & Le, Q. V. (2020). Adversarial examples improve image recognition. In Proceedings of the IEEE/CVF Conference on Computer Vision and Pattern Recognition (pp. 819-828).
[2] Wong et al. Fast is Better than Free: Revisiting Adversarial Training, ICLR 2020
[3] Rice et al., Overfitting in adversarially robust deep learning, ICML 2020, https://arxiv.org/abs/2002.11569
[4] Gowal et al., An Alternative Surrogate Loss for PGD-based Adversarial Testing, https://arxiv.org/pdf/1910.09338.pdf
[5] Croce et al., Reliable Evaluation of Adversarial Robustness with an Ensemble of Diverse Parameter-free Attacks, ICML 2020

In summary, although the observations presented in this work are interesting, further analysis and thorough evaluations are required to justify the claims made in this paper.

Expecting the authors to address the following during rebuttal period:

- Please address and clarify the cons as listed above.
- Could the authors please provide additional details on how the finetuning is performed in Section 3.1?
- In Figure1, could the authors clarify the quantity along the x-axis? If it represents iterations, or epochs?
- Could the authors clarify how the plot for Adv-Train is obtained (in relation to the Std-BN finetune plot)? How were the updates performed on the Adv-Train model to obtain this plot?
- Given that the parameters of the first BN layer are presented in Figure1, since the convolutional layer that occurs before this BN layer is frozen, it is expected that Plots 1(a) and 1(b) are identical for the two cases, and does not offer additional insight.
- Could the authors comment on the behaviour of BN parameters in deeper layers of the network when the same fine-tuning experiment is performed?
- In Figure2, we observe that very few feature maps are indeed changed. Could the authors comment on this? Also, could the authors clarify which layer was used to obtain these figures?
- In Algorithm 1, could the authors clarify if the number p is sampled uniformly from the [0,1] range?
- Further, it is not clear which set of BN parameters is used in the crafting of the PGD attack in the proposed algorithm. This small detail is likely to cause dramatic changes in the final outcome of training. As currently presented, it appears as though the BN parameters corresponding to standard training are used in this step.
- Could the authors show an ablation experiment where the BN parameters corresponding to RobMask are used for attack generation to understand this better?
- Could the authors clarify if the network parameters $\theta$ are also adversarially updated (along with W and W’) using the loss on the perturbed sample $x+\delta$ as presently shown for all experiments? If so, could the authors clarify why the accuracy for eps=0 in Table4 differs from that shown in Table3 (5-6% difference)?
- Also, it is not completely clear from the algorithm if BN parameters of all layers in the network are updated, or if it is restricted to some specific layer.
- Could the authors share details used for adversarial training (optimizer, learning rate schedule, number of epochs, validation split, use of early stopping)? These factors play a crucial role in the final robust accuracy achieved and in the trade-off with clean accuracy as well, as often the model obtained at the last epoch of training achieves lower adversarial accuracy compared to intermediate epochs.

########################## Update after rebuttal ##########################

I thank the authors for their detailed response; several concerns have been addressed in the rebuttal. I would encourage the authors to use commonly used practices to improve the robust performance of models in Tables 5 and 6. The use of early-stopping [3] can significantly boost the robust performance, and produce models with better clean accuracy than is presently reported for Adv. Training. I would like to update the score to 5 based on the author's response. Aside from the robust evaluation, the improvement in clean accuracy is of a relatively smaller magnitude given the disproportionate increase in training requirements. Thus, I have not further increased the score.

[3] Rice et al., Overfitting in adversarially robust deep learning, ICML 2020, https://arxiv.org/abs/2002.11569

---

> ### Author Response · Authors · 2020-11-21
> **Response to AnonReviewer2 (Part 1)**
>
> Thanks for your insightful and detailed comments. Below, we provide detailed responses to your questions.  However, I want to first clarify our contributions. Our main contribution is in two folds. First, we discover an interesting phenomenon that changing the batch normalization itself with all other weights fixed can control the tradeoff between adversarial robustness and generalization. Second, we formulate this finding into adversarial masking and propose RobMask to boost the generalization performance. As a side contribution, it also brings an additional benefit that robustness-accuracy trade-off is improved.
>
> Q1: About the novelty compared with AdvProp [1].
>
> A1: AdvProp [1] has an assumption that the trade-off is caused by distribution mismatch, i.e., adversarial examples and clean images are drawn from two different domains, therefore training exclusively on one domain cannot well transfer to the other. In Section 3.1, we show the running mean and variance of adversarial and clean examples are almost the same, so their hypothesis is questionable. Instead, we show it is the rescaling parameter together with ReLU function, which is what we call “adversarial masking”, causing the trade-off. Also, as stated, AdvProp cannot use multiple perturbation strengths in their training, which is nontrivial to solve.  Essentially, our method starts from a different standpoint, and we propose a more general method to solve the problem.
>
> Q2: Results of setting k=3 and what the different w_k’s would represent?.
>
> A2: For k=3, we add another dimension so that we could achieve a slightly better accuracy 94.67% for 20 epochs and a comparable accuracy 95.88% for 100 epochs training on ResNet-18 in CIFAR10 dataset.
> Each w_k denotes a “basis” and all the BN parameters for each different epsilon are based on a linear combination of these bases. In general, we found k=3 does not significantly improve the performance, so we still recommend using k=2.
>
> We also want to emphasize that RobMask with k=2 is different from Advprop. RobMask considers a series of epsilon, and uses different batch norm (a linear combination of base w_k’s) for each epsilon. In comparison, Advprop only uses a fixed two batch norms, one for clean and one for epsilon perturbed data. Therefore RobMask significantly outperforms Advprop in the experiments.
>
> Q3: About the intuition of the trade-off improvement.
>
> A3: RobMask incorporates a range of perturbations instead of just one. Empirically this leads to a better generalization than Advprop, which brings a better trade-off.
>
>
> Q4: About the training time in DenseNet.
>
> A4: The training time of AdvProp and RobMask is almost the same. Since both RobMask and AdvProp use two forward and one backward pass, they have almost identical training time per epoch. For example, in DenseNet-121, it takes around 870 seconds for Advprop and around 890 seconds for RobMask.  i.e., our model did not take much longer to train.  For the standard training, it is clearly shown in Table 3 that even the 100th epoch result is worse than RobMask at 20 epochs. Giving more epochs will not improve the performance of standard training a lot, you could also refer to https://github.com/kuangliu/pytorch-cifar, where they reported that the peak performance for densenet (with standard training) is 95.04%.
>
> Q5: The clean accuracy on the adv trained model.
>
> A5: We use our own implementation on adversarial trained models, and do not use a lot of tricks that proposed in [2,3]. Also, to be noted, the number listed in [2] is for PreAct ResNet18, which is a different architecture. However, to be noted, the improvement on trade-off is considered as a side contribution of RobMask and the main contribution is we discover an interesting phenomenon that changing the batch normalization itself with all other weights fixed can control the tradeoff between adversarial robustness and generalization and propose RobMask to boost the generalization performance.
>
> Q6: Could the authors provide additional results for k=3 or k=4?
>
> A6: Same as Q2.
>
> Q7: Could the authors clarify the exact BN parameters used in final evaluation on clean and adversarial samples?
>
> A7: Since there are still discrepancies between different u_i, therefore, if we want to obtain the best clean performance, we use p=0. If we want to obtain the best robustness performance, p=1 for the adversarial samples is used for the evaluation. We will make this clear in the revision.
>
>
> Q8: Evaluation on PGD-100 step attack alone is not sufficient。
>
> A8: Thanks for the suggestion. We add the Autoattack results in Table 5. Using Autoattack, robust accuracy of every method is decreasing by 2-3% while the gap between every methods keeps around the same.

---

> ### Author Response · Authors · 2020-11-21
> **Response to AnonReviewer2 (Part 2)**
>
> Q9: Could the authors show results where the maximum constraint is set to 10/255?
>
> A9: Thanks for the suggestion. We add it in Table 5. Yes, we could see the 8/255’s robust accuracy is improved from 41.7% to 41.69%; however, it sacrifices some clean accuracy.
>
> Q10: Additional details on how the finetuning is performed in Section 3.1.
>
> A10: As stated in the paper, we start from an adversarial/clean trained model. Freezing other parameters except all batch normalization layers, we fine-tuned the network with the same optimizer, learning rate, learning rate scheduler and number of epochs. We will make this clear.
>
> Q11: X-axis in Figure1
>
> A11: The x-axis is across the dimension. The first batch norm as shown in Figure 1 has dims=64 so we have x indexed from 0 to 63.
>
> Q12: Could the authors clarify how the plot for Adv-Train is obtained (in relation to the Std-BN finetune plot)? How were the updates performed on the Adv-Train model to obtain this plot?
>
> A12:  As stated in Figure 1’s title, we conduct standard fine-tuning of only the BN parameters of an adversarially-trained network.
>
> Q13: Since the convolutional layer that occurs before this BN layer is frozen, it is expected that Plots 1(a) and 1(b) are identical.
>
> A13: Plot 1(a) and 1(b) are based on the running mean and variances trained by only fine-tuning on adversarial examples or clean examples. If the intuition in AdvProp [1] is true, they should be completely different, since they are in different distributions. However, we found only the rescaling weight has changed dramatically while running mean and variances stay unchanged.
>
> Q14: BN in deeper layer.
>
> A14: We show the experiment in Table 9 in Appendix. It could be clearly observed that the rescaling weight has the most significant difference across all the batch normalization layers’ parameters. However, since the deep layer’s mean and variance would be affected by the shallow layers’ rescaling parameter, the result on the deeper layer couldn’t disentangle the effect between normalization and rescaling because it is mixed.
>
> Q15: “very few feature maps changed In Figure 2. which layer was used to obtain Figure 2?”
>
> A15: In figure 2, we only marked several changes. If you look closely, 16/64=25% feature maps have changed dramatically. In fact, all the feature maps change to some extent, however, we could only observe some in the figure. We extract the feature maps after the first batch normalization and ReLu layer.
>
> Q16: the number p is sampled uniformly from the [0,1] range In Algorithm 1?
>
> A16: Yes, you are correct. We will make this clear.
>
> Q17: “which set of BN parameters is used in the crafting of the PGD attack in the proposed algorithm.”
>
> A17: Sorry for the confusion, we use the forward’s BN parameters to generate adversarial attack and to perform adversarial training so the BN parameters are not corresponding to standard training.
>
> Q18: Could the authors show an ablation experiment where the BN parameters corresponding to RobMask are used for attack generation to understand this better?”
>
> A18: Thanks for the suggestion. However, we use the BN parameters on the attack generalization same as training. Could you clarify how the experiments should be done?
>
> Q19: Could the authors clarify if the network parameters θ are also adversarially updated (along with W and W’) using the loss on the perturbed sample x+δ”
>
> A19: Yes, you are correct. We will make this clear.
>
> Q20: could the authors clarify why the accuracy for eps=0 in Table4 differs from that shown in Table3 (5-6% difference)?”
>
> A20: During attack, we use the mask corresponding to p=1 which has some performance drop since it has a different mask with p=0.
>
> Q21: Also, it is not completely clear from the algorithm if BN parameters of all layers in the network are updated, or if it is restricted to some specific layer.
>
> A21: All BN layers parameters are updated.
>
> Q22: Could the authors share details used for adversarial training (optimizer, learning rate schedule, number of epochs, validation split, use of early stopping)?
>
> A22: We use SGD optimizer with the cosine learning rate scheduling and trained by 100 epochs. We set the learning rate=0.1. We didn’t use early stop and use the standard validation split same as advprop and RobMask.
>
>
> Please let us know if you have additional questions, and we are happy to answer them. Thank you!
>
> [1] Ye, Chengxi, et al. "Network deconvolution." arXiv preprint arXiv:1905.11926 (2019).

---

> ### Author Response · Authors · 2020-11-24
> **Thanks for the update. However, we want to address the improvement.**
>
> Thanks for the great suggestion on the adversarial training details and we keep updating it in further revision. However, we still want to address our improvement on the generalization, especially for the clean data. As mentioned in the previous response, the training time of AdvProp[2] and RobMask is almost the same. While kept the running time equally, RobMask has achieved a much more significant improvement (+1 to 2 percent) over normal training and (+0.5 to 1.5 percent) over AdvProp. To our best knowledge, the improvement over generalization around 1% to 2% should be considered as significant. Some works such as Advprop and Network Deconvolution [1] achieving performance improvements around 1% are considered as significant.
>
> Please let us know if you have any other concerns or disagreements with our responses and claims. Thanks!
>
>
>
>
> [1] Ye, Chengxi, et al. "Network deconvolution." ICLR 2020.
> [2] Xie, C., Tan, M., Gong, B., Wang, J., Yuille, A. L., & Le, Q. V. (2020). Adversarial examples improve image recognition. In Proceedings of the IEEE/CVF Conference on Computer Vision and Pattern Recognition (pp. 819-828).

---

### Official Review · AnonReviewer3 · 2020-10-28
**An interesting extension of AdvProp with limited evidence of practicality**

**Rating:** 6
**Confidence:** 4

**Review:**

## Overview

The paper focuses on the generalization issue with adversarial training that various work has recently demonstrated. The paper studies the role of batch normalization (BN) in adversarial robustness and generalizability. The authors single out the rescaling operator in BN to significantly impact the clean and robustness trade-off in CNNs.  They then introduce Robust Masking (Rob-Mask), which is shares similarities to the CVPR2020 paper by Xie et al. (2020). Xie et al. use an auxiliary BN in adversarial training, which uses different batch normalization parameters for adversarial samples, to improve the generalizability of CNNs. Still, the authors clearly state the differences between Rob-Mask and AdvProp.

## Contributions

The contributions of the paper are as follows:

1. Showing the effect of BN (and, more specifically, the scale parameter of BN together with ReLU) as adversarial masking.

   a. Authors show that *adversarial fine-tuning* of only the BN parameters of a *vanilla-trained* network provides some adversarial robustness, although at the trade-off losing test accuracy.

   b. Authors show that *standard fine-tuning* of only the BN parameters of an *adversarially-trained* network increases the network's generalizability, although at the trade-off losing robustness.
2. Showing that interpolating between the BN parameters in Contribution 1 provides a smooth trade-off between generalizability and robustness.

3. Devising an approach for utilizing different perturbation strengths for model training. The authors build on their Contribution 2 and propose $k$ basic (or better to say principle) rescaling parameters, the linear combination of which leads to a rescaling parameter.

4. Providing a short yet informative, ablation study to show the effectiveness of Contribution
5. Showing experimental benefits over AdvProp on CIFAR10 and CIFAR100 datasets.

Contribution 3 turns AdvProp into a particular case of RobMask.  In fact, Xie et al. (2020) mention in their paper that "a more general usage of multiple BNs will be further explored in future works," which seems to be the inspiration behind this paper.

## Weaknesses

1. The main limiting factor for the impact of this paper is the experiments. The paper only reports performance on CIFAR10 and CIFAR100. Given that the paper can be considered an extension/improvement over AdvProp, it is desirable to have similar largescale experiments in Xie et al. (2020) on ImageNet and its variations. A head-to-head comparison with the experiments in Xie et al. (2020) would provide a clearer picture to show the proposed method's power.
2. Regarding the practicality of the approach, I am missing a computational analysis of the approach to compare it against BN and AdvProp, e.g., it would be great if the authors provided a head-to-head comparison of training curves. Does your method take much longer to train?
3. How many times did you run each experiment? What are the standard deviations in Table 3 (and other tables)? Providing this information, at least in the supplementary materials, could clarify your results' statistical significance.

## Questions and comments for the authors

1. The notation $\gamma_i$ is used both for BN's scaling parameter and for the learning rate, which turns the equations hard to follow.
2. On the bottom of page 7, you wrote: "It is because both AdvProp and Adversarial training models are trained with adversarial examples generated with $\epsilon= 8/255$, while our methods use a random perturbation where $\epsilon_{max}=8/255$."  The term "random perturbation" is misleading here, as I believe you are also using PGD attack, but the adversarial perturbation's strength is randomized. Is that correct?
3. Please refer to Weaknesses 2.
4. I don't find Figure 2 informative at all. I suggest that the authors remove the figure and use the space to address the raised concerns.

## Evaluation logic

I find the paper an interesting extension of the CVPR2020 paper by Xie et al. However, the paper's experimental section does not provide enough information to the reader to see the concrete benefit of the proposed method in training a large scale CNNs. I think the paper could significantly benefit from a more extensive experimental setting. Given the limited novelty and lack of concrete evidence of practicality, I score the paper as a 5.

## Post rebuttal evaluation

I thank the authors for providing answers to the raised questions and providing further experiments. Regarding Figure 3, I suggest that the authors provide accuracy as a function of wallclock instead of epochs currently reported in the paper. As a result of the authors' responses, I increase my score to 6.

---

> ### Author Response · Authors · 2020-11-21
> **Response to AnonReviewer3**
>
> Thanks for your insightful comments. Below, we provide detailed responses to your questions.
>
> Q1: About ImageNet results and head-to-head comparison with AdvProp.
>
> A1: Thanks for the great suggestion. However, we couldn’t find the official training code for AdvProp. Also, it is pretty difficult and time-consuming to train EfficientNet in our own servers without TPU support. Therefore, it is very difficult for us to perform a head-to-head comparison with AdvProp, though we acknowledge that it is a great suggestion. As a result, we were only able to conduct comparisons with AdvProp on CIFAR10/CIFAR100 in the submission.
> To address your concern, recently we have found a pytorch implementation at https://github.com/tingxueronghua/pytorch-classification-advprop for training ResNet on ImageNet. Due to time limit, we have added the ResNet-18 ImageNet experiments in Table 3 in the revision, showing that the proposed method still outperforms AdvProp. Specifically, we achieve +0.38% accuracy improvement over standard training, while AdvProp only achieves +0.03% accuracy improvement based on our experiments.
>
>
> Q2: About head-to-head comparison of training curves and training time.
>
> A2: We have added Figure 4 in the revised paper. Since both RobMask and AdvProp use two forward and one backward pass, they have almost identical training time per epoch. For example, in DenseNet-121, it takes around 870 seconds for Advprop and around 890 seconds for RobMask.  i.e., our model did not take much longer to train.
>
>
> Q3: How many times did you run each experiment?”
>
> A3: We run each experiment three times. We have added the standard deviation in Table 8 in the Appendix.
>
>
> Q4: About the notation \gamma_i and Figure 2.
>
> A4: Thanks for the suggestions. We have changed the learning rate to \eta_i, and moved Figure 2 to Appendix.
>
>
> Q5: I believe you are also using PGD attack, but the adversarial perturbation's strength is randomized.
>
> A5: Yes, we set p to be sampled uniformly from [0,1], so the adversarial perturbation’s strength is randomized from 0 to \epsilon_max.

---

### Official Review · AnonReviewer4 · 2020-10-29
**Nice empirical observation on the trade-off between generalization and robustness**

**Rating:** 7
**Confidence:** 2

**Review:**

The paper observes that the rescaling operation in the batch normalization layer and the ReLU activation learn to select different features for standard and adversarial trainings. The authors call this effect "Adversarial Masking." Based on this observation, the authors then propose Rob-Mask that achieves good standard and adversarial accuracies at the same time.

The trade-off between standard and adversarial accuracies are an important problem in adversarial machine learning. The findings in this paper are really interesting and deserve further investigation from the research community.
My biggest concern is that, since it is an empirical paper without rigorous guarantees, I would like to see the experiments on more datasets. The experiments in this paper are only on CIFAR-10 and CIFAR-100, which are similar datasets. Does the finding still hold in other datasets like ImageNet?
Also, there are many models that do not use Batch Normalization, but the standard accuracies also drop when doing adversarial training. The proposed hypothesis cannot explain that. One ablation test that I would like to see is whether fine-tunning a single convolutional layer will have similar effect as fine-tunning the BN layer done in this paper.

---

> ### Author Response · Authors · 2020-11-21
> **Response to AnonReviewer4**
>
> Thanks for your encouraging and insightful comments. Below, we provide detailed responses to your questions.
>
> Q1: About the ImageNet results.
>
> A1: Thanks for the great suggestion. However, we couldn’t find the official training code for AdvProp. Also, it is pretty difficult and time-consuming to train EfficientNet in our own servers without TPU support. Therefore, it is very difficult for us to perform a head-to-head comparison with AdvProp, though we acknowledge that it is a great suggestion. As a result, we were only able to conduct comparisons with Advprop on CIFAR10/CIFAR100 in the submission.
> To address your concern, recently we have found a pytorch implementation at https://github.com/tingxueronghua/pytorch-classification-advprop for training ResNet on ImageNet. Due to time-limit, we have added the ResNet-18 ImageNet experiments in Table 3 in the revision, showing that the proposed method still outperforms AdvProp. Specifically, we achieve +0.38% accuracy improvement over standard training, while AdvProp only achieves +0.03% accuracy improvement based on our experiments.
>
>
> Q2: The proposed hypothesis cannot explain why the standard accuracies also drop when performing adversarial training on models that do not use Batch Normalization.
>
> A2: Thanks for the great question. Since Batch normalization is widely used in nearly all state-of-art neural networks, our analysis could be applied directly. However, we haven’t talked about the robustness of feature extraction in the convolution layer and we leave it as a future work to discuss the tradeoff if there is no batch norm.
>
>
> Q3: About fine-tuning a single convolution layer.
>
> A3: Thanks for the suggestion. We have conducted the experiments only fine-tuning a single convolution layer while freezing others. When starting from a standard trained model and performing adversarial fine-tuning, the clean accuracy drops from 91.97% to 46.46% and robust accuracy only improves marginally from 0.0% to 1%.
> When starting from an adversarial trained model and performing standard training, the clean accuracy increase from 78.47% to 81.3% and robust accuracy drops significantly from 48.56% to 4.58%. Therefore, We shows if we only fine-tune on a single convolution layer, we will achieve a very bad tradeoff between clean and robust errors.
> However, our main focus here is based on that we can learn robust features from a vanilla standard-trained model.  And we leave the robustness of feature extraction in the convolution layer for the future work.

---

### Official Review · AnonReviewer1 · 2020-11-09
**Simple way to turn adversarial training as a regularizer**

**Rating:** 7
**Confidence:** 4

**Review:**

Summary:
This paper follows the direction of previous work AdvProp (Xie et al. (2020)) and aims to use adversarial training as a regularizer to improve the network generalization on clean data. The authors analyze that the different rescaling operation in the batch normalization layer along with ReLU acts as feature masking/selection layer, which can control the trade-off between adversarial robustness and clean data performance. Unlike AdvProp, which uses different batch normalization layer for clean images and adversarial images at a specific perturbation strength, here the authors propose a technique called RobMask that adapts the rescaling parameters of batch normalization based on the perturbation strength during training. The authors show that such adapting technique is more effective than using different batch normalization layers (as in AdvProp) for each perturbation strength and thus improves the clean data performance on CIFAR10/100.

Strengths:
+ Motivation and analysis is clear.
+ Discussed its differences to previous works.
+ Proposed technique is simple, easy to adapt in the existing setup of adversarial training and addresses the limitations of the previous work AdvProp.
+ Evaluation is carried out on CIFAR10/100 across different network architectures: ResNet-18, DenseNet-121, Preact ResNet18, ResNeXt-29.
+ Results on CIFAR-10/100 using four different deep network architectures suggest that this work improves clean data performance than the baselines: standard network training and AdvProp.

Weaknesses:
-	The authors claim about well-balanced robustness trade-off using their method and also claim that their major objective is only to improve network generalization on clean data. There is a little ambiguity regarding the major contribution of this paper. The authors can make this point more clear.
-	Isn’t the hypothesis that is stated as “new” in this work already discussed in AdvProp i.e. using different batch normalization for clean and adversarial images improves network generalization, which in turn draw the conclusion that rescaling operation of batch norm could control the robustness and generalization trade-off. Why this hypothesis considered as “new” then ?
-	The two learned adversarial maskings discussed in section 3.2, it is not clear how they are generated.
-	Results demonstrate that the proposed approach improves generalization but the performance gain is minimal (only 1%-2%) and not so significant compared to the baselines.
Minor point:
-	I understand that the major objective of this work is to improve performance on clean images but not the adversarial robustness. The results demonstrate higher robustness against PGD based adversarial attacks with perturbation strength lower than 8/255 is interesting but not of practical importance since the method requires perturbation strength as an additional input and very specific to PGD based attack. I wouldn’t consider this as major weakness since it is not the primary objective of this work.

Final thoughts:
The proposed method is clearly motivated. Although the performance gains on network generalization are minimal compared to the baselines, this work cleverly addressed the limitations of previous work and extend it with simple modifications. I tend to accept this paper. However, I suggest the authors to also consider the evaluations carried out in AdvProp (Xie et al. (2020)) to improve the significance of their work.

---

> ### Author Response · Authors · 2020-11-21
> **Response to AnonReviewer1**
>
> Thanks for your encouraging and insightful comments. Below, we provide detailed responses to your questions.
>
> Q1: About the major contribution.
>
> A1: Sorry for the confusion. Our main contribution is in two folds. First, we discover an interesting phenomenon that changing the batch normalization itself with all other weights fixed can control the tradeoff between adversarial robustness and generalization. Second, we formulate this finding into adversarial masking and propose RobMask to boost the generalization performance. As a side contribution, it also brings an additional benefit that robustness-accuracy trade-off is improved.
>
>
> Q2: Compared with Advprop, why is the hypothesis in this paper considered as “new”?
>
> A2: As stated in Section 4 (Connection with Advprop), Advprop has an assumption that the trade-off is caused by distribution mismatch, i.e., adversarial examples and clean images are drawn from two different domains, therefore training exclusively on one domain cannot well transfer to the other. In Section 3.1, we show the running mean and variance of adversarial and clean examples are almost the same, so the “distribution mismatch” hypothesis in the AdvProp paper is not completely true. Instead, it is the rescaling parameter together with ReLU function, i.e., what we call “adversarial masking”, causes the trade-off.
>
> Further,  by exploiting the low rank structure, we propose RobMask to incorporate different perturbation strengths for model training, instead of just one.  RobMask boosts generalization on clean data and achieves a better trade-off between robust and natural accuracy over Advprop.
>
>
> Q3:It is not clear how the adversarial maskings in Section 3.2 are generated.
>
> A3: It is constructed by the first batch normalization (BN) layer’s rescaling parameter with the later ReLU function after the first convolution layer. We obtained them by only fine-tuning the BN layer, the same as the fine-tuning experiment.
>
>
> Q4: The performance gain is minimal.
>
> A4:  To our best knowledge, the improvement over generalization around 1% to 2% should be considered as significant.  For example, works such as Advprop and Network Deconvolution [1] achieving performance improvements around 1% are considered as significant.
>
>
> Q5: Consider the evaluations carried out in AdvProp (Xie et al. (2020)).
>
> A5: Thanks for the great suggestion. However, we couldn’t find the official training code for AdvProp. Also, it is pretty difficult and time-consuming to train EfficientNet in our own servers without TPU support. Therefore, it is very difficult for us to perform a head-to-head comparison with AdvProp, though we acknowledge that it is a great suggestion. As a result, we were only able to conduct comparisons with Advprop on CIFAR10/CIFAR100 in the submission.
> To address your concern, recently we have found a pytorch implementation at https://github.com/tingxueronghua/pytorch-classification-advprop for training ResNet on ImageNet. Due to time-limit, we have added the ResNet-18 ImageNet experiments in Table 3 in the revision, showing that the proposed method still outperforms AdvProp. Specifically, we achieve +0.38% accuracy improvement over standard training, while AdvProp only achieves +0.03% accuracy improvement based on our experiments.
>
> [1] Ye, Chengxi, et al. "Network deconvolution." arXiv preprint arXiv:1905.11926 (2019).

---

### Public Comment · ~Jingfeng_Zhang1 · 2020-11-11
**Some similar papers about the topic "tradeoff between robustness and accuracy."**

Dear Authors,

It is enjoying reading your paper.
The paper's topic is "Understanding Robustness Trade-off for Generalization."

Recently there are some similar papers (below) also discussing this tradeoff. Could I know the (dis)similarities between theirs and yours? It would be good if you could at least discuss some :)

[1]  Understanding and mitigating the tradeoff between robustness and accuracy. In ICML, 2020
[2] Attacks which do not kill training make adversarial learning stronger. In ICML, 2020
[3] A closer look at accuracy vs. robustness. In NeurIPS 2020

---

> ### Author Response · Authors · 2020-11-21
> **Our main contribution is understanding batch norm in the trade-off and improving generalization, which is different with the listed relevant papers.**
>
> Thanks for your interests. [1] introduces unlabeled data to achieve a better trade-off. [2] uses early stop PGD to get least adversarial data in the adversarial training. [3] theoretically suggests that it is possible to learn classifiers both robust and highly accurate on real image data. None of the aforementioned methods could achieve a better generalization on clean data. Therefore, different with [1,2,3],  our main contribution is that we study an interesting phenomenon that changing the batch normalization itself with all other weights fixed can control the tradeoff between adversarial robustness and generalization. Therefore, we formulate this finding into adversarial masking and propose RobMask to boost the generalization performance. We will add more discussion in the later revision.

---

### Decision · Program_Chairs · 2021-01-07
**Final Decision**

**Decision:**

Reject

**Comment:**

The reviews were a bit mixed, with some concerns on the incremental nature of this work, which the AC concurs (after independently going through both the submission and Xie et al 2020). In a nutshell, the main contribution on the authors' side appears to be a simple linear interpolation of two masks so that it is possible to leverage attacks with varying strengths. Other claimed contributions are not substantiated. In particular:

(a) Fig 1 and its conclusion are a bit disturbing. It is suggested that the authors back up their claim with more empirical and theoretical evidence. For example, why can one conclude from the same mean and variance that there is no distribution mismatch between clean and adversarial examples (as claimed in Xie et al)? If the two sources have similar distribution, why is there a sharp difference in gamma for the two? When one claims different results from previous work, due diligence is required. For instance, did the authors reproduce the mean and variance on the same architecture and dataset of Xie et al? How about other BN layers (in addition to the first one)? How to explain the difference in gamma? The fact that you are using different masks for different epsilon is an indication that their distributions are probably different. The authors mentioned the joint effect between gamma and relu activation, which could be potentially insightful. However, this is a bit speculative in its current presentation. How about an ablation study with leaky relu or tanh/sigmoid? Without these careful comparisons, these claimed contributions are not appropriate to publish in their current form.

(b) As the reviewers pointed out, why BN, after all for models that do not use BN they still suffer from adversarial examples? Even when we restrict to models that use BN, why replicating BN for different sources helps generalization? Here, an excellent experiment point is to compare to fine-tuning or replicating other layers in the network. During rebuttal, the authors only tried to fine-tune ONE convolution layer and quickly concluded its ineffectiveness. Note that in contrast the authors fine-tuned ALL BN layers. How about all convolution layers, some pooling layers, the last softmax layer? These experiments could help us understand if there is some magic in BN. Or maybe it is just more convenient to fine-tune BN because of its small number of parameters? In any case these experiments would largely strengthen the findings of this work.

(c) As pointed out by the reviewers, Xie et al hinted at the advantage of using multiple BN masks and the authors proposed to linearly interpolate the masks. In the ablation study, what if we increase the number of BN masks in Xie et al, say we discretize p into 11 values p = {0, 0.1, ..., 0.9, 1} and have 1 mask for each value? Here an interesting experiment is to compare K = 11 (basically AdaProp) with smaller K (such as 2 or 5). The authors seemed to suggest that a larger K does not seem to help, which would be clarified through the preceding experiment (and perhaps more). Note that using a uniformly random p is equivalent as adversarial training with a weaker (and varied) attack, and the better tradeoffs shown in the experimental section (e.g. Table 3) are perhaps expected.

(d) As pointed out by the reviewers, a head to head comparison against AdaProp (preferably with more masks) is desirable. The authors mentioned some difficulty in conducting this experiment fully. If it is only the software side, maybe check the sources here:
https://paperswithcode.com/paper/adversarial-examples-improve-image

(e) Finally, a minor point: Algorithm 1 with k=2 and p=1 does not reduce to AdaProp as one will only train on adversarial examples and ignore all clean samples?

In the end this submission appears to be a bit incremental. However, the authors are strongly suggested to follow the reviewers' comments to further polish their work and address the concerns above. With proper revision this work can eventually become a solid contribution on top of AdaProp.